# Changes in miR-124-1, miR-212, miR-132, miR-134, and miR-155 Expression Patterns after 7,12-Dimethylbenz(a)anthracene Treatment in CBA/Ca Mice

**DOI:** 10.3390/cells11061020

**Published:** 2022-03-17

**Authors:** Andras Tomesz, Laszlo Szabo, Richard Molnar, Arpad Deutsch, Richard Darago, Bence L. Raposa, Nowrasteh Ghodratollah, Timea Varjas, Balazs Nemeth, Zsuzsanna Orsos, Eva Pozsgai, Jozsef L. Szentpeteri, Ferenc Budan, Istvan Kiss

**Affiliations:** 1Doctoral School of Health Sciences, Faculty of Health Sciences, University of Pécs, 7624 Pécs, Hungary; laszlo.szabo.pte@gmail.com (L.S.); richard.molnar.pte@gmail.com (R.M.); deutscharpad@gmail.com (A.D.); daragorichard@gmail.com (R.D.); raposa.bence@gmail.com (B.L.R.); 2Department of Public Health Medicine, Medical School, University of Pécs, 7624 Pécs, Hungary; taytakh@yahoo.com (N.G.); vtimi_68@yahoo.com (T.V.); nem_bal2@hotmail.com (B.N.); zsuzsa.orsos@aok.pte.hu (Z.O.); pozsgay83@gmail.com (E.P.); istvan.kiss@aok.pte.hu (I.K.); 3Institute of Transdisciplinary Discoveries, Medical School, University of Pécs, 7624 Pécs, Hungary; 4Institute of Physiology, Medical School, University of Pécs, 7624 Pécs, Hungary

**Keywords:** miRNA, carcinogen, 7,12-dimethylbenz(a)anthracene, miR-132, miR-212, miR-124-1, miR-155, miR-134

## Abstract

Specific gene and miRNA expression patterns are potential early biomarkers of harmful environmental carcinogen exposures. The aim of our research was to develop an assay panel by using several miRNAs for the rapid screening of potential carcinogens. The expression changes of miR-124-1, miR-212, miR-132, miR-134, and miR-155 were examined in the spleen, liver, and kidneys of CBA/Ca mice, following the 20 mg/bwkg intraperitoneal 7,12-dimethylbenz(a)anthracene (DMBA) treatment. After 24 h RNA was isolated, the miRNA expressions were analyzed by a real-time polymerase chain reaction and compared to a non-treated control. DMBA induced significant changes in the expression of miR-134, miR-132, and miR-124-1 in all examined organs in female mice. Thus, miR-134, miR-132, and miR-124-1 were found to be suitable biomarkers for the rapid screening of potential chemical carcinogens and presumably to monitor the protective effects of chemopreventive agents.

## 1. Introduction

Cancer has become a major public health problem worldwide. Due to the increasing trendline of lifestyle-related risk factors, as well as current trend of population ageing, cancer morbidity is predicted to reach 28.4 million cases worldwide by 2040, which is a 47% increase compared to the estimated number of cases in 2020 [1]. According to the World Health Organization (WHO), however, about half of all malignant tumors are preventable [2], and early diagnosis—in general—increases the chances of successful treatment. Consequently, the timely identification of harmful exogenic substances that may play a part in tumor development, the early detection of tumors, along with the continuous development of more accurate diagnostic and more effective therapeutic procedures are all important factors in the prevention and successful treatment of cancer.

Specific gene expression patterns, such as increased expression of *HA-RAS* or *C-MYC* oncogenes, are early biomarkers of harmful environmental exposures at the different stages of tumorigenesis and represent the underlying mechanisms of the carcinogenic process [3,4,5,6]. Moreover, previously, we found that miR-9-3 and mTORC1 can be used as biomarkers for early detection of DMBA-induced damages in female mice [7].

MicroRNAs (miRNAs) are highly conserved RNA molecules of 19 to 22 nucleotides in length that regulate approximately 30–50% of human gene expression [8,9]. They bind to the 3′ untranslated region of partially or fully complementary messenger RNAs (mRNAs) and thereby inhibit their translation and ultimately the protein synthesis [10]. Since miRNAs are stable in vivo and show a mostly altered expression pattern in malignantly transformed cells [11,12], distinct miRNA patterns can be observed following exposure to carcinogenic agents [13].

The carcinogenesis-related regulatory functions of miRNAs can be highly diverse, ranging from oncogenic to tumor suppressing effects, depending on the nature of the genes they regulate [14]. The complexity of their regulation is supported by the fact that some miRNAs can have both oncogenic and tumor suppressor properties, because a miRNA can be partially or fully complementary to several different mRNAs. For example, the miR-125 as a tumor suppressor reduces the antiapoptotic effect of the *MUC1* and *Bcl-2* genes [15,16] and suppresses vascular endothelial growth factor (VEGF) expression, thereby inhibiting tumor proliferation and metastasis formation [17]. However, miR-125 also acts as a tumor promoter by regulating the tumor suppressor protein TP53INP1 (Tumor protein P53 induced nuclear protein 1), and, as a result, stimulates cell proliferation and migration [15].

MiR-124-1, miR-212, miR-132, miR-134, and miR-155 are implicated in the regulation of oncogenes, tumor suppressor genes, cell growth, proliferation, invasion, migration, metastasis, and apoptosis through various signaling pathways [18,19,20]. Moreover, these miRNAs have been found to show altered expression levels in a variety of malignant cancer cells [18,19,21]. MiR-124 acts as a tumor suppressor by inhibiting the translation of the cell cycle regulator cyclin dependent kinase 6 (CDK6) and by reducing the phosphorylation of the retinoblastoma protein (Rb) [19]. Furthermore, loss of miR-124 function by hypermethylation is frequently found in tumor cells [18]. Proteins regulated by miR-212 affect cell proliferation and apoptosis as well as processes involved in tumorigenesis, such as invasion and metastasis formation [22]. For example, as a target of miR-212, the transcription factor SOX4 has been found to be involved in the invasion, migration, and metastasis of various tumors [23]. The miR-132 has both tumor-specific oncogenic and tumor suppressor properties. Its oncogenic effect has been observed in pancreatic carcinoma (PC) by inhibiting the *Phosphatase and tensin homolog (PTEN)* tumor suppressor gene [24]. However, in non-small-cell lung carcinoma (NSCLC) miR-132 has been shown to inhibit migration and invasion of cells [25], thus indicating a tumor suppressor characteristic. The miR-134 increases the expression of *P21* and *P57* CDKIs and represses the expression of cyclin D1, cyclin D2, and cyclin-dependent kinase 4 (CDK4) proteins, which are promoters of the cell cycle and are overexpressed in many tumors. Thus, miR-134, as a tumor suppressor, reduces cell proliferation [26] and enhances apoptosis [27]. MiR-155 is significantly overexpressed in cervical cancer (CEC) tissues compared to normal tissues [28].

Many chemical compounds, with the ability to cause cancer in humans, have been identified. 7,12-dimethylbenz(a)anthracene (DMBA) is one of the most potent mutagenic and carcinogenic environmental polycyclic aromatic hydrocarbons [29,30]. Polycyclic aromatic hydrocarbons are formed during imperfect burning of organic compounds; thus, for example, cigarette smoke and car exhaust gases are characteristic and significant sources of DMBA exposure [31]. DMBA is activated by cyto-chrome P450-metabolizing enzymes [32,33]. The resulting metabolites can alkylate the DNA or other cellular macromolecules, which explains its initiator and promoter effects [6]. In addition, DMBA enhances the formation of reactive oxygen species (ROS) [34], and this effect also contributes to cellular dysfunction and altered behavior [35]. DMBA is often used in animal experiments as an initiator carcinogenic agent [36], leading to the formation of tumors histologically similar to human cancers [37]. The expression of certain genes (e.g., *H-RAS*, *C-MYC*, and *P53*) and miRNAs (e.g., miR-21 and miR-146a) is altered 24 to 48 h after treatment with DMBA; thus, these expression changes can be considered as early biomarkers of the carcinogenic process [38,39].

Increased expression levels of certain miRNAs, namely miR-128b, miR-152, miR-125b, miR-205, miR-27a, miR-146a, miR-222, miR-23a, miR-24, miR-150, etc., were represented in lung cancer, while the expression levels of miR-29a, miR-221, miR-223, miR-25, miR-92, miR-99a, etc., were increased in colorectal cancer tissues [40]. Moreover, it has been reported that patients with papillary thyroid carcinomas showed increased levels of miR-221, miR-222, and miR-146 in the tumors [41]. Thus, the role of miRNAs as early epigenetic biomarkers in various tumors has been confirmed [42]. Therefore, we hypothesized that altered expression patterns of certain miRNAs may be biomarkers of early damage caused by the carcinogenic agent, DMBA.

The aim of our research was to develop an assay panel for the rapid screening of potential carcinogens or chemopreventive agents. Earlier, we investigated the expression of miR-330, miR-29a, miR-9-1, miR-9-3, and mTORC1 after treatment with DMBA in vivo and found that miR-9-3 and mTORC1 expression in female mice could be suitable biomarkers for rapid identification for possible carcinogenic effects [7]. Following up on our previous study, to expand the options for the biomarkers in the assay panel, we analyzed the expression of miR-124-1, miR-212, miR-132, miR-134, and miR-155 after DMBA exposure in CBA/Ca mice spleen, liver, and kidneys. The aim of the present study was to investigate whether any or some of these miRNAs could be used as potential biomarkers of early carcinogenic damage as well as to elucidate the probable role of miRNAs in the process of tumorigenesis.

## 2. Materials and Methods

### 2.1. Animal Treatment

Two groups of CBA/Ca mice were used in our study. Both control and DMBA-treated groups consisted of 12 (six males and six females) 6–8 weeks old mice. Both the treated and the control groups received 0.1 mL of corn oil (Sigma-Aldrich, St. Louis, MO, USA) intraperitoneally, but in the treated group, 20 mg/bwkg DMBA was dissolved in the corn oil. After 24 h of DMBA exposure, the mice were euthanized, cervical dislocation was performed, and the livers, kidneys, and spleens were removed. Total cellular RNA level was determined as described below.

In accordance with the guidelines concerning laboratory animals, mice received humane care. The experiment was approved by Regional Animal Ethical Committee Pécs and conducted according to the current ethical regulations (ethical permission no.: BA02/2000-79/2017).

### 2.2. Isolation of Total RNA

Total cellular RNA was isolated using TRIZOL reagent, according to the manufacturer’s instructions (Thermo Fisher Scientific, Waltham, MA, USA). The RNA quality was determined by NanoDrop absorption photometry. Only RNA fractions with A > 2.0 at 260/280 nm were used for the reverse transcription polymerase chain reaction process.

### 2.3. Reverse Transcription Polymerase Chain Reaction (RT-PCR)

Following the manufacturer’s instructions, the one-step PCR, including reverse transcription and target amplification, was performed in a 96-well plate using Kapa SYBR FAST One-step RTQCR kit (Kapa Biosystems, Wilmington, MA, USA) on a LightCycler 480 qPCR platform.

The temperature program was set as follows: 5 min incubation at 42 °C and 3 min incubation at 95 °C. Then, 45 cycles (95 °C—5 s, 56 °C—15 s, and 72 °C—5 s) were performed, and at the end of each cycle, a fluorescent reading was made. Each run was performed with melting curve analysis (95 °C—5 s, 65 °C—60 s, and 97 °C ∞) to confirm the amplification specificity. The reaction mixture used was the following: 10 μL KAPA SYBR FASTqPCR Master Mix, 0.4 μL KAPA RT Mix, 0.4 μL dUTP, 0.4 μL primers, and 5 μL miRNA template supplemented with sterile double-distilled water to a total volume of 20 μL.

Primer sequences for the examined miRNAs (miR-124-1, miR-212, miR-132, miR-134, and miR-155) and the internal control gene (mouse U6) are summarized in Table 1.

Primers were synthetized by Integrated DNA Technologies (Integrated DNA Technologies Inc., Coralville, IA, USA), and sequences were taken from previous publications [43,44].

### 2.4. Calculations and Statistical Analysis

Relative miRNA expression levels were calculated and compared using the 2^−ΔΔCT^ method. During the statistical analysis for the testing the distribution of results, we used the Kolmogorov–Smirnov test. To compare means, we used the Levene’s type T-probe. For calculations and analysis, IBM SPSS 21 (International Business Machines Corporation, Armonk, NY, USA) statistical software was used. We determined the level of statistical significance at *p* < 0.05.

Differences in miRNA expressions were expressed on the figures as percentages of the respective control group.

## 3. Results

### 3.1. Changes in miRNA Expression in the Liver 24 h after DMBA Treatment

A total of 24 h after DMBA treatment, there was a highly significant (*p* < 0.001) increase in the expressions of miR-134, miR-132, miR-124-1, miR-212, and miR-155 in the liver of DMBA-treated female mice compared to the control group (Figure 1A). In contrast, only miR-124-1 decreased expression was significant (*p* < 0.05) among the tested miRNAs in the liver of treated males (Figure 1B).

### 3.2. Changes in miRNA Expression in the Spleen 24 h after DMBA Treatment

In the spleens of female mice, DMBA induced a highly significant (*p* < 0.001) rise in the expressions of miR-124-1, miR-132, and miR-134 compared to the control group (Figure 2A). A highly significant (*p* < 0.001) increase in miR-132, miR-212, and miR-155 expression could also be observed in the spleens of males after DMBA treatment, as shown in Figure 2B.

### 3.3. Changes in miRNA Expression in the Kidneys 24 h after DMBA Treatment

Figure 3A illustrates that DMBA treatment resulted highly significant (*p* < 0.001) increase in the expression of miR-134, miR-132, miR-124-1, and significant (*p* < 0.05) increase in the expression of miR-212 in the kidneys of female mice. On the other hand, miR-134 and miR-124-1 showed a highly significantly decreased expression, and miR-155 demonstrated a highly significantly increased expression in the kidneys of male mice in response to DMBA compared to untreated male mice (Figure 3B).

## 4. Discussion

We found a highly significant increase in the expressions of miR-134, miR-132, and miR-124-1 following exposure to DMBA in all the examined organs of female mice; however, less-marked and sometimes opposite changes in miRNA expression could be detected in DMBA-treated males. These results indicate that miR-134, miR-132, and miR-124-1 could be potential biomarkers of early carcinogenic damage in females and that gender-specific hormonal differences affect miRNA expression patterns caused by carcinogenic factors, significantly. DMBA, a well-known carcinogen, affects various members of the cell signaling pathways, including miRNA expressions, which, in turn, influence the activation or inhibition of other key elements of signal transduction pathways.

DMBA induces a dose-dependent increase in transforming growth factor beta 1 (TGF-β1) levels [45] and the higher expression of TGF-β1 increases miR-132 expression [46]. However, miR-132 blocks the DMBA-activated TGF β1/Smad2/3 signaling [47], suggesting that there is a mutual feedback between TGF-β1 and miR-132. In addition, miR-132 regulates liver tumor cell proliferation, apoptosis, migration, and invasion by suppressing the DMBA-induced transcription factor SRY-related HMG-box (SOX) [48,49]. It is also known that miR-132 represses the Akt/mTOR signaling pathway, which is activated by DMBA and involved in promoting tumor formation [50]. The significant increases in miR-132 expression, observed in our study, may be an early response to DMBA-induced signaling processes such as SOX, TGF-β1, or mTOR activation. Our present results suggest that miR-132 cannot be used as a biomarker in males. However, a highly significant increase in miR-132 expression was found in females, correlating with literature data [45,46]. Thus, miR-132 could be used as a gender-specific biomarker in female mice.

Both in vitro and in vivo studies confirm the complex role of miR-212 in carcinogenesis [51,52]. MiR-212 inhibits cell proliferation, migration, and invasion in renal cell carcinoma (RCC) and exerts tumor suppressor effects by downregulating T-box transcription factor TBX15 (TBX15) [51,52]. In the present study, the results regarding miR-212 expression in the liver were opposite in the two genders (decreasing in males and increasing in females), which supports the mechanism of tumor development via the Forkhead Box A1 (FOXA1) signaling pathway, while miR-212 suppresses the growth of hepatocellular carcinoma (HCC) by downregulating the expression of the *FOXA1* gene [53]. This effect is gender-specific because FOXA1 is involved in both estrogen and androgen signaling, acting as a central regulator of sexual dimorphism in liver tumors, which are more common in males [54]. An increase in miR-212 expression in females reduces the protective effect of estrogen by suppressing FOXA1, whereas in males, the reduced miR-212 levels increase FOXA1 expression, which favors the hepatic tumor-promoting effect of androgen signaling [52,53,54]. These observations are supported by the report that the suppression of FOXA1 resulted in more and larger tumor cells in females and reduced tumor growth in males [55]. This highlights the interactions between sex hormones and signal transducers in liver tumors, but further studies are required to understand the exact role of miRNAs, for example miR-212, in the regulation of these processes. Interestingly, miR-212 expression in the spleen of female mice did not show any significant changes after DMBA treatment; therefore, miR-212 could only be used as a biomarker limited to the liver organ at most.

According to an earlier investigation, increased expression of miR-155 expression could be observed in vivo in the liver, spleen, lung, and kidneys 3 and 6 h after DMBA treatment [39]. In accordance with this, our results showed a significant increase in miR-155 expression in the spleen and kidneys of males and the liver of females 24 h after DMBA exposure. However, unidirectional, positive changes were found only in the liver from the studied organs in both sexes, although the changes were not significant in males, which are explained by the miR-155 upregulating effect of estrogen [56]. This effect may be due to the metabolic activation of DMBA by cytochrome P450 (CYP) enzymes, mainly cytochrome P450 1A1 (CYP1A1) and 1B1 (CYP1B1), which are typically present in the liver with high enzyme activity, and thus the effect of DMBA on miR-155 expression may be more rapid and dominant in this tissue type [57]. It is also known that miR-155 is overexpressed in liver injury, and miR-155 exerts a protective effect by suppressing nuclear factor-kappa B (NF-kB) signaling [58]. In addition, miR-155 has also been identified as an oncomiR in various types of human tumors, which inhibits the activity of tumor suppressor targets such as TP53INP1 or the von Hippel–Lindau (VHL) tumor suppressor protein [59,60]. The data in the literature as well as in our study highlight the complex function of miR-155 in carcinogenesis and indicate the need for further studies to fully understand the observed changes and the mechanisms influencing miR-155 levels. Therefore, based on the present results, as well as the mentioned multifarious factors, miR-155 cannot be used as a robust early biomarker in our target assay panel.

The CASC3 protein is a direct target of miR-124, through which miR-124 is capable of inactivating the p38-MAPK, JNK, or ERK signaling pathways, thereby inhibiting cell proliferation [61]. The tumor suppressor effect of miR-124 was also observed through the inhibition of cell cycle-related cyclin D1, cyclin-dependent kinase 2 (CDK2), cyclin-dependent kinase 4 (CDK4), and cyclin-dependent kinase 6 (CDK6) [62,63]. Despite these previous findings, the expected increase in miR-124-1 expression following DMBA exposure was only observed in the female group, with highly significant increases in all three examined organs. DMBA activates the signal transducer and activator of transcription 3 (STAT3) transcription factor [64], which has a repressive effect on miR-124 [65]. The STAT3-inducing effect was also observed after testosterone treatment, which may explain the downregulation of miR-124-1 in the male group [66]. Therefore, miR-124-1 was found to be potential biomarker only in females.

Strong expression of miR-134 leads to reduced expression of the *C-MYC* oncogene, and the Cyclin E and Cyclin D1 cell cycle regulatory proteins. Furthermore, the increased expression of miR-134 coincides with growing cyclin-dependent kinase inhibitor 1B (p27) levels, which also inhibits the cell cycle [67,68]. It also acts as a tumor suppressor by regulating intracellular signal transduction pathways such as the RAS/MAPK/ERK or the RAS/PI3K/AKT signaling pathways that stimulate cell proliferation and invasion and inhibit apoptosis [69,70]. According to literature data, there is a significantly reduced expression of miR-134 in renal cell carcinoma [69]. This observation correlates with our findings that there was a significant decrease in miR-134 expression in the kidney tissue of the male mice 24 h after DMBA exposure. However, a not-significant reduction was also observed in the other tested organs, so the effect of miR-134 as a tumor suppressor was weaker in male mice. In contrast, DMBA treatment caused a significant increase in miR-134 expression after 24 h in the kidneys, liver, and spleen of the females. Upstream regulators of miR-134, such as the NF-kB transcription factor, whose aberrant activation has been reported in malignant tumor cell lines, may underlie the clear and significant sex differences [71]. DMBA increases the expression of NF-kB [72], resulting in a stronger repression of miR-134 [73], which may explain the observed decreases in miR-134 expression in male mice in our study. In female mice, however, the estrogen receptor (ER) inhibits NF-kB activation [74], thus blocking its repressive effect on miR-134 and leading to an increased miR-134 expression.

Increased expression of miR-124 also has a silencing effect on NF-kB by regulating TRAF6 [75], thereby reducing the NF-kB-induced miR-134 inhibition. This is also reflected in our findings (Figure 4) since we found very similar expression patterns of miR-134 and miR-124-1. Thus, it is conceivable that miR-124-1 regulates miR-134 expression through the TRAF6/NF-kB pathway [73,75].

Considering the effects of sex-specific hormonal differences and the statistically significant expressions of miR-134 in the examined organs, miR-134 was found to be a usable biomarker only in females.

Figure 5 shows the summary of the affected relevant signaling pathways and the role of the expression changes of the investigated miRNAs (miR-134, miR-132, miR-124-1, miR-212, and miR-155) 24 h after DMBA treatment.

## 5. Conclusions

We investigated the expression of miR-134, miR-132, miR-124-1, miR-212, and miR-155 as well as in a previous study other miRNAs and the *mTORC1* after in vivo DMBA treatment to establish a biomarker system of early carcinogenic damages [7].

In our present study, we observed statistically significant increases in expression of miR-134, miR-132, and miR-124-1 in all examined organs of female mice in contrast to the males, mainly due to different effects of sex hormones, based upon literature. Summarizing our present and previous results, miR-9-3, miR-124-1, miR-132, and miR-134 miRNAs, as well as the *mTORC1* gene, were usable for detecting carcinogen exposure in female CBA/Ca mice. In addition to examining the early effects of DMBA, the model can also be used to study chemopreventive effects, by the simultaneous use of supposed chemopreventive agents and DMBA, measuring to what extent these agents are able to reduce/prevent the DMBA-induced miRNA and epigenetic changes after 24 h [76,77,78,79,80].

The complexity of the molecular processes involved in carcinogenesis makes it necessary to extend the test systems to the widest possible range of signaling pathways and their regulators. The miR-9-3, miR-124-1, miR-132, and miR-134 miRNAs and the *mTORC1* gene are involved in signaling pathways such as RAS/MAPK/ERK, RAS/PI3K/AKT, p38 MAPK, or JNK, which typically show higher activity in the early phase of carcinogenesis. In addition, these four miRNAs regulate the synthesis of various growth factors, growth factor receptors, signal transduction proteins, transcription factors, cell cycle regulators, tumor suppressor genes, inflammatory mediators, and, indirectly, even additional miRNAs that play key roles in proliferative activity and/or carcinogenesis [76]. With our panel—due to the broad involvement of signaling pathways and regulatory proteins concerned in carcinogenesis—we can detect early molecular processes induced by chemical carcinogens and the protective effects of chemopreventive agents with higher sensitivity and complexity than conventional gene expression models. Compared to single-element gene or miRNA expression assays, this robust analytical strategy gives an opportunity for more efficient qualitative and quantitative detection of putative carcinogen exposure. Furthermore, in the future, the model can be further optimized by including additional miRNAs and genes and by exploring possible differences in their importance.

Applied in practice, the five biomarker-based assay panel designed to predict carcinogenesis could provide a valuable tool for further investigation of chemopreventive and/or complementary therapeutic tumor suppressor compounds, thus opening new opportunities for reducing tumor incidence and mortality by using them as a tool for primary, secondary, and tertiary prevention.

## Figures and Tables

**Figure 1 cells-11-01020-f001:**
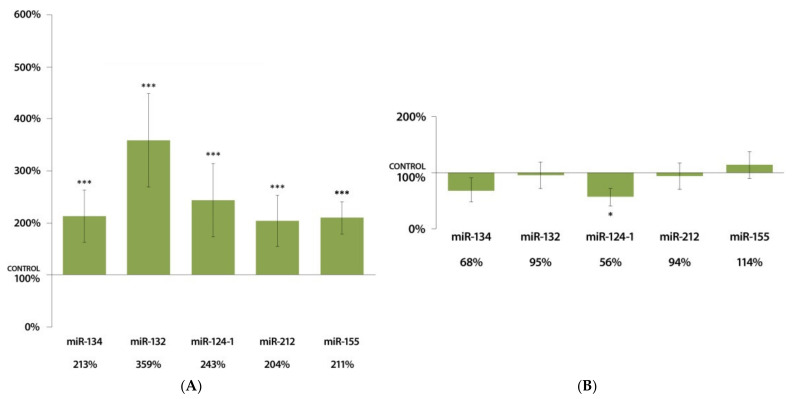
Expression results of miR-134, miR-132, miR-124-1, miR-212, and miR-155 in the liver of (**A**) female and (**B**) male CBA/Ca mice 24 h after DMBA treatment, compared to non-treated mice (100%) (* *p* < 0.05; *** *p* < 0.001).

**Figure 2 cells-11-01020-f002:**
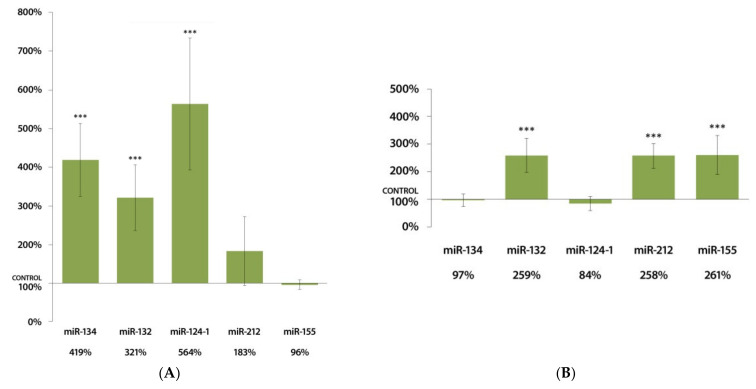
Expression results of miR-134, miR-132, miR-124-1, miR-212, and miR-155 in the spleen of (**A**) female and (**B**) male CBA/Ca mice 24 h after DMBA treatment, compared to non-treated mice (100%) (*** *p* < 0.001).

**Figure 3 cells-11-01020-f003:**
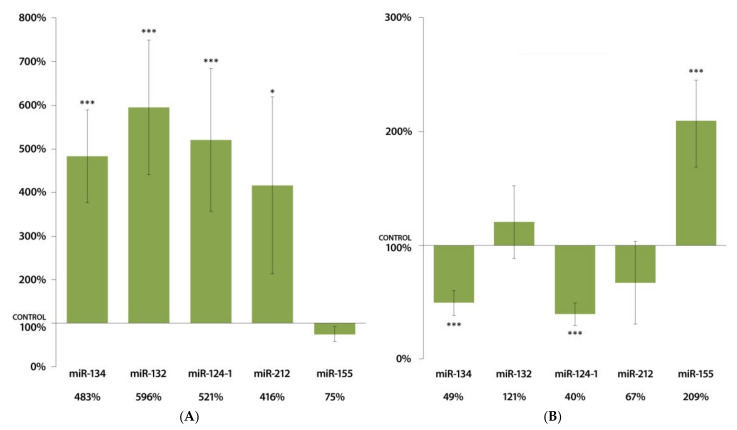
Expression results of miR-134, miR-132, miR-124-1, miR-212, and miR-155 in the kidneys of (**A**) female and (**B**) male CBA/Ca mice 24 h after DMBA treatment, compared to non-treated mice (100%) (* *p* < 0.05; *** *p* < 0.001).

**Figure 4 cells-11-01020-f004:**
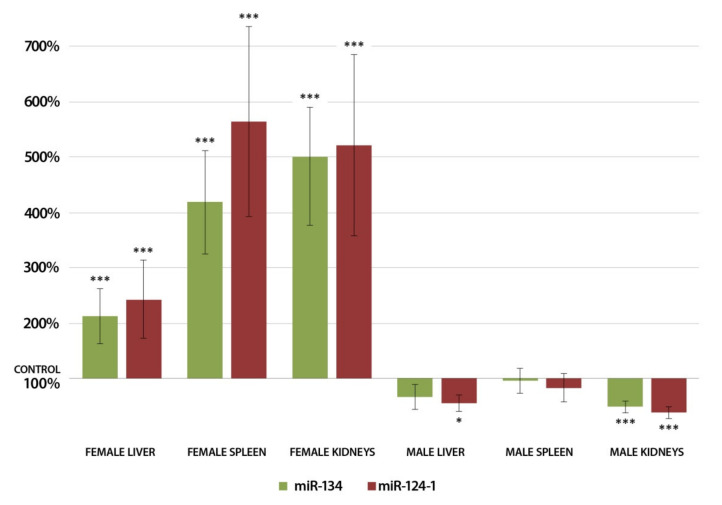
Expression results of miR-134 and miR-124-1 in the liver, kidneys, and spleen of CBA/Ca mice 24 h after DMBA treatment, compared to non-treated mice (100%) (* *p* < 0.05; *** *p* < 0.001).

**Figure 5 cells-11-01020-f005:**
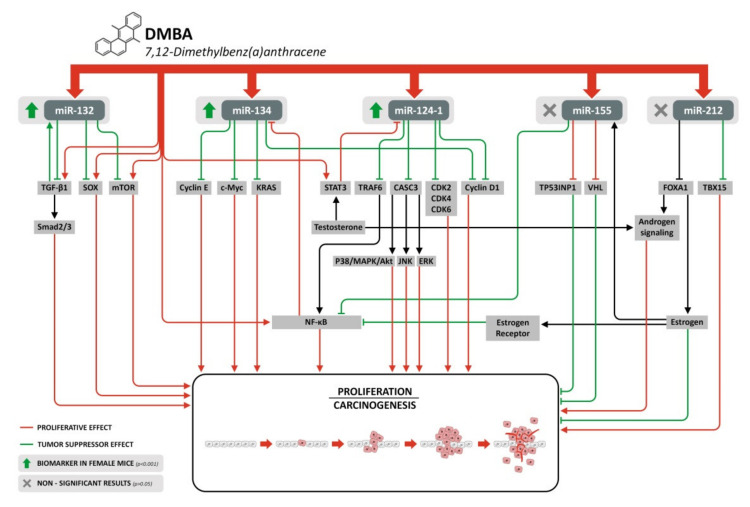
Summary of the affected relevant signaling pathways and the role of the expression changes of the investigated miRNAs (miR-134, miR-132, miR-124-1, miR-212, and miR-155) 24 h after DMBA treatment.

**Table 1 cells-11-01020-t001:** The used primer sequences for the examined miRNAs (miR-124-1, miR-212, miR-132, miR-134, and miR-155) and the internal control gene (mouse U6).

miRNA	FORWARD	REVERSE
miR-134	TGTGACTGGTTGACCAGAGG	GTGACTAGGTGGCCCACAG
miR-132	ACCGTGGCTTTCGATTGTTA	CGACCATGGCTGTAGACTGTT
miR-124-1	TCTCTCTCCGTGTTCACAGC	ACCGCGTGCCTTAATTGTAT
miR-212	GGCACCTTGGCTCTAGACTG	GCCGTGACTGGAGACTGTTA
miR-155	GACTGTTAATGCTAATCGTGATAG	GTGCAGGGTCCGAGGTATTC
mouse U6	CGCTTCGGCAGCACATATAC	TTCACGAATTTGCGTGTCAT

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
