# Peer review of "Changes in miR-124-1, miR-212, miR-132, miR-134, and miR-155 Expression Patterns after 7,12-Dimethylbenz(a)anthracene Treatment in CBA/Ca Mice"

_cells, 2022, doi:10.3390/cells11061020_

Round 1

Reviewer 1 Report

In general, the paper is interesting, however, it needs to be improved before being published. In introduction section the importance of DMBA should be added or included.  

In line 83 and 84 you mention “The miR-134 increases the expression of P21, that represses the expression of cyclin D1, cyclin D2 and Cyclin-dependent kinase 4 (CDK4) proteins, which are promoters of the cell cycle and are overexpressed in many tumors.” However, miR-134 increase diminished the expression of cyclin D1, cyclin D2 and Cyclin-dependent kinase 4 (CDK4) proteins independently of p21 expression. I recommend to rephase this sentence.

In line 95 The references 33 and 34 do not support the expression of H-Ras, C-myc and p53. I recommend to added references that support the expression of H-Ras, C-myc and p53 in response to DMBA.

Regarding the design of primers, it should be clarified because its design it is referenced to publications 38 and 39, however, in these papers they evaluated miRNA expression by Northern blot. Additionally, I will like to see the raw data of RT-PCR as well as a scheme of its amplification.

Author Response

Dear Reviewer,

The authors would like to thank for the thorough review of our paper and for all the valuable comments. Hereby please find our answers to the questions:

„In introduction section the importance of DMBA should be added or included.”

According to the Reviewer’s suggestions, in the revised manuscript we expanded the introduction with information on DMBA, including it’s mode of action, major sources of exposure, etc. (new references have been added to this part as well).

„In line 83 and 84 you mention “The miR-134 increases the expression of P21, that represses the expression of cyclin D1, cyclin D2 and Cyclin-dependent kinase 4 (CDK4) proteins, which are promoters of the cell cycle and are overexpressed in many tumors.” However, miR-134 increase diminished the expression of cyclin D1, cyclin D2 and Cyclin-dependent kinase 4 (CDK4) proteins independently of p21 expression. I recommend to rephase this sentence.”

Thank you very much for noticing the contradiction. The sentence was corrected as follows: „ The miR-134 increases the expression of P21 and P57 CDKIs, and represses the expression of cyclin D1, cyclin D2 and Cyclin-dependent kinase 4 (CDK4) proteins, which are promoters of the cell cycle and are overexpressed in many tumors.”

„In line 95 The references 33 and 34 do not support the expression of H-Ras, C-myc and p53. I recommend to added references that support the expression of H-Ras, C-myc and p53 in response to DMBA.”

Thank you for the comment. Reference 33 has been changed to an appropriate reference in the revised manuscript.

Regarding the design of primers, it should be clarified because its design it is referenced to publications 38 and 39, however, in these papers they evaluated miRNA expression by Northern blot. Additionally, I will like to see the raw data of RT-PCR as well as a scheme of its amplification.

Thank you for the comment. The primer sequences, taken from reference 39, however, are not for Northern blot. This reference uses two methods (RT-PCR and Northern-blot), and Table S1 contains the promer sequences for the RT-PCR (the Northern-blot sequences for some other RNAs are given in Table S2). Several other papers use these primer sequences in RT-PCR studies as well [1-6].

During the analysis we used the standard method, so that we normalized our data to the expression of U6 endogeneous reference gene, and relative to the non-treated groups (all the results are expressed as fold increase, compared to the non-treated groups).

Unfortunately, we are unable to send the amplification curves, since during a period of Covid-epidemic our PCR machine was used for PCR-testing, and when we received it back, all the data have been deleted, together with patient’s data. In case it is necessary, we can provide the Excel files with the existing data.

Thank you again for the comments and review, and we hope that our answer addressed all the required issues.

  1. Wang, S. Lu, J. Jiang, X. Jia, X. Dong, and P. Bu: Hsa-microRNA-101 suppresses migration and invasion by targeting Rac1 in thyroid cancer cells. Oncol Lett. 2014 Oct; 8(4): 1815–1821.
  2. Ahmadvand et al.: Over expression of circulating miR-155 predicts prognosis in diffuse large Bcell lymphoma. Leukemia Research 70 (2018) 45–48.
  3. Saad et al.: Regulation of Brain DNA Methylation Factors and of the Orexinergic System by Cocaine and Food Self-Administration. Mol Neurobiol. 2019 56(8):5315-5331.
  4. Uchida et al.: Characterization of the vulnerability to repeated stress in Fischer 344 rats: possible involvement of microRNA-mediated down-regulation of the glucocorticoid receptor. European Journal of Neuroscience, 27: 2250–2261, 2008
  5. Higuchi et al.: Hippocampal MicroRNA-124 Enhances Chronic Stress Resilience in Mice. J Neurosci. 2016 Jul 6; 36(27): 7253–7267.
  6. Zhang, M. Huang, Z. Cao, J. Qi, Z. Qiu, L-Y. Chiang: MeCP2 plays an analgesic role in pain transmission through regulating CREB / miR-132 pathway. Mol Pain. 2015; 11: 19.

Reviewer 2 Report

In the article, “Changes in miR-124-1, miR-212, miR-132, miR-134 and miR-155 expression patterns after 7,12-dimethylbenz(a)anthracene treatment in CBA/Ca mice”,  the authors Tomesz et al describe a small study whereby the expression patterns of miR were interrogated following DBA exposure in CBA/Ca mice. The goal of this study is to develop a screening panel of biomarkers that could be used to detect early exposure to carcinogenic chemical that might lead to tumor development. This is an interesting, albeit small, study that poses several interesting potential findings. Before this manuscript can be published, however, a few questions need to be answered.

  • How complementary are the miRNA primer sequences and the miRNAs they are target to that of humans?
  • Trizol-based extraction methods of total RNA has been shown to impact total RNA yield.  Also, repeated freeze-thaw cycles of samples have also led to inconsistency in mRNA levels due to degradation. I am curious whether the authors took measures to guard against both from happening and if they could describe those methods.   

Author Response

Dear Reviewer,

The authors would like to thank for the effort to read and review our manuscript. Please allow us to answer your questions in the following section:

“How complementary are the miRNA primer sequences and the miRNAs they are target to that of humans?”

There is often a high level of homology among microRNAs deriving from different species (even if these species are quite far from each other, such as C. elegans, D. melanogaster and humans), which suggests that these regulatory sequences are evolutionally highly conserved [1,2]. This does not imply a complete and perfect homology between all human and mouse microRNAs, the homology is rather seen between the seed sequences of human and mouse miRNAs (however, sometimes complete homology can be seen) [4-7]. Overall, there is an approximately 60% homology between mouse and human miRNAs [3]. Since the homology is less strong for the complete pri-miRNA sequence and for its flanking regions, the human and mouse miRNA primer sequences are typically different. Our knowledge on this field is far from complete, but numerous papers assign identical functions to several miRNAs in mice than to their human counterparts, and their disturbed functions are often found in the same mouse and human tumors [8-10].

Altogether, while certain differences exist between the mouse and human miRNAs, mouse models from the studied point of view, can be considered as good animal models for developing exposure biomarkers.

“Trizol-based extraction methods of total RNA has been shown to impact total RNA yield.  Also, repeated freeze-thaw cycles of samples have also led to inconsistency in mRNA levels due to degradation. I am curious whether the authors took measures to guard against both from happening and if they could describe those methods.”

Repeated freezing and thawing is a well-known problem for studying nucleic acids. While miRNAs are in general more stable than mRNAs, the proble still exists in relation to miRNAs as well. We routinely use in our lab the simplest, but in our opinion, the most effective measure against this: the samples are divided into several parts so that we do not have to reuse (i.e., freeze and later thaw again) the thawed sample. While single-use aliquots definitely require more storage space, the number of studies we perform in our lab is not extremely high, so we can keep up with storage space.

The other part of the question is related to yield:

We had used the classic phenol-choloroform method for nucleic acid isolation in our lab for approximately 15 years, so we had experience on working with DNA and with the more sensitive RNA. We took all the measures necessary to preserve the integrity of RNA (such as ensuring an RNAse free environment, using DEPC treated water, taking care of temperature, storage conditions, contamination, etc.). When we switched to Trizol, we tried different protocols, and we settled down at ThermoFisher (both for the Trizol reagent and for the protocol). Our current study uses mouse organs as starting point for the RNA isolation, and therefore yield was not a key issue. We naturally always measure the concentration and purity of the total RNA, but we do not routinely measure the proportion of the miRNAs in the total RNA. While there are RNA isolation methods with definitely higher yield, the amount of nucleic acids, including miRNAs was always appropriate for the subsequent qPCR reactions. Instead, purity was the most important criterion, the A260/280 ratio is taken very seriously. We tried magnetic beads based isolation (we still have a Roche MagnaLyser in our lab), it is undoubtedly more comfortable, but because of cost-effectivity we still prefer the Trizol method. Besides strictly keeping the isolation protocol, applying the mentioned principles, according to our experience one thing might be mentioned specifically, related to the yield – the homogenization. We noticed that it may significantly affect the yield, primarily the length of homogenization. We do not exceed 30 seconds in order to prevent destruction/degradation of nucleic acids.

Thank you again for your review, and we hope we were able to addresse all the required issues.

  1. Ibáñez-Ventoso, M. Vora and M. Driscoll: Sequence Relationships among C. elegans, D. melanogaster and Human microRNAs Highlight the Extensive Conservation of microRNAs in Biology. PLoS One. 2008; 3(7): e2818.
  2. Tal et al: MicroRNAs control neurobehavioral development and function in zebrafish. The FASEB Journal. 2012; 26(4):1452-61.
  3. Pal and AL. Kasinski: Animal models to study microRNA function. Adv Cancer Res. 2017; 135: 53–118.
  4. Sempere, S. Freemantle, I. Pitha-Rowe, E. Moss, E. Dmitrovsky and V. Ambros: Expression profiling of mammalian microRNAs uncovers a subset of brain-expressed microRNAs with possible roles in murine and human neuronal differentiation. Genome Biology Vol 5, Article number: R13 (2004)
  5. Weber: New human and mouse microRNA genes found by homology search. FEBS J. 2005 Jan;272(1):59-73.
  6. Ichiyama and C. Dong: The role of miR-183 cluster in immunity. Cancer Letters 443 (2019) 108–114.
  7. Jiang et al.: MiR-20b Down-Regulates Intestinal Ferroportin Expression In Vitro and In Vivo. Cells 2019, 8(10), 1135.
  8. Peng and CM. Croce: The role of MicroRNAs in human cancer. Signal Transduction and Targeted Therapy (2016) 1, 15004.
  9. Chen et al: Integrated Analysis of Mouse and Human Gastric Neoplasms Identifies Conserved microRNA Networks in Gastric Carcinogenesis. Gastroenterology 2019;156:1127–1139.
  10. Liu et al: MicroRNA-31 functions as an oncogenic microRNA in mouse and human lung cancer cells by repressing specific tumor suppressors. J Clin Invest. 2010 Apr;120(4):1298-309.